# Transcriptome Analysis and GC-MS Profiling of Key Fatty Acid Biosynthesis Genes in *Akebia trifoliata* (Thunb.) Koidz Seeds

**DOI:** 10.3390/biology11060855

**Published:** 2022-06-03

**Authors:** Yicheng Zhong, Yunlei Zhao, Yue Wang, Juan Niu, Zhimin Sun, Jianhua Chen, Mingbao Luan

**Affiliations:** 1Key Laboratory of Stem-Fiber Biomass and Engineering Microbiology, Ministry of Agriculture, Institute of Bast Fiber Crops, Chinese Academy of Agricultural Sciences, Changsha 410205, China; 82101195049@caas.cn (Y.Z.); 82101205107@caas.cn (Y.W.); 82101181085@caas.cn (J.N.); sunzhimin@caas.cn (Z.S.); 2Cotton Research Institute, Chinese Academy of Agricultural Sciences, Anyang 455000, China; zhaoyunlei@caas.cn

**Keywords:** *Akebia trifoliata*, seed oil, fatty acid biosynthesis

## Abstract

**Simple Summary:**

Plant oil is an important renewable energy substance, and *A. trifoliata* seeds are of value in this regard. *A. trifoliata* fruits have many seeds with high oil content, but research progress on *A. trifoliata* seed oil is slow. Fatty acid biosynthesis is the most important factor affecting plant oil content. Therefore, analysis of the key genes for fatty acid biosynthesis is beneficial for breeding *A. trifoliata* varieties with high oil content. Here, we report changes in seed oil and key oil biosynthesis genes in the growth period of *A. trifoliata* based on transcriptome analysis. We found that the development of *A. trifoliata* seeds and fruits was not synchronized, and when the fruit was ripe, the seed oil content was not the highest. With the development of *A. trifoliata* seeds, linoleic and oleic acid content was found to decrease and increase, respectively. Subsequently, several key genes for oil biosynthesis in *A. trifoliata* were identified. These results further our understanding of the mechanism underlying oil biosynthesis in *A. trifoliata* seeds.

**Abstract:**

*Akebia trifoliata* (Thunb.) Koidz is an important Chinese medicinal and economic crop. Its seeds, which are rich in fatty acids, are usually discarded. As of now, *A. trifoliata* lipid biosynthesis pathways and genes have not been clearly described. In this work, we found that seed and fruit development of *A. trifoliata* were not synchronized, and that when the fruit was ripe, seed oil content was not at its highest. As seeds developed, linoleic and oleic acid content was found to decrease and increase, respectively. RNA sequencing yielded 108.45 GB of clean reads from 15 cDNA libraries, containing 8756 differentially expressed genes. We identified 65 unigenes associated with lipid biosynthesis, including fatty acid and triacylglycerol biosynthesis. The 65 unigenes were mapped to the *A. trifoliata* lipid synthesis pathway. There were 20 *AtrFAD* family members in *A. trifoliata*, which could be divided into four sub-groups with the highest number of *AtrSADs*. Our study revealed the dynamic changes in *A. trifoliata* seed oil content and composition during its growth period and provides large-scale and comprehensive transcriptome data of *A. trifoliata* seeds. These findings provide a basis for the improvement of *A. trifoliata* seed oil yield and quality.

## 1. Introduction

With global advancements in industrialization, there is an increasing demand for fossil fuel-derived energy; however, fossil fuels are a non-renewable energy source. Solving the energy crisis and achieving sustainable development are issues that must be urgently addressed [1]. Fatty acids (FAs) are widely distributed in plants and have been considered a renewable energy source to replace petroleum [2]. Plants mostly contain C16–C20 FAs, which can serve as efficient energy sources. Plant FAs have been used in many fields, including the food, chemical, and energy industries [3,4]. FAs also constitute an important energy source for humans, and edible FAs are mainly derived from plants [5]. The main difference between edible and industrial oils is in their composition; if the palmitic acid and stearic acid content of plant oil is too high, then it will be considered unsuitable as edible oil. Presently, the main industrial oil crops globally include *Trachycarpus fortunei* and *Vernicia fordii* [6,7].

*Akebia trifoliata* (Thunb.) Koidz is widely distributed in China, Japan, and Russia. In China, it has been used in traditional Chinese medicine for over two thousand years [8]. Furthermore, it was included in the most recent edition (10th edition) of the European Pharmacopoeia [9]. In recent years, *A. trifoliata* fruits have been widely accepted as having health benefits, and there has been an increase in its planting area in China. However, this fruit contains many seeds and has a thick pericarp, resulting in low edibility, which has affected its cultivation [10,11]. Its seeds and pericarp are usually discarded, resulting in a waste of resources. However, the seeds contain a lot of oil and proteins and therefore have high processing value [12,13].

At present, the advancement in research on *A. trifoliata* seed oil (ASO) is slow and mainly focuses on its components and extraction. ASO is composed mainly of C16 and C18 FAs, such as palmitic acid and oleic acid [14]. Zhou et al. showed that the Soxhlet extraction method was the most effective method for ASO extraction [15], and Zhong et al. showed that a near-infrared spectroscopy model could accurately predict ASO content [12]. However, a few studies have been carried out on ASO biosynthetic pathways [16]. Generally, plant oil biosynthesis pathways can be divided into three parts: FA synthesis, triacylglycerol synthesis, and oil body synthesis. Acetyl-CoA carboxylase, FA desaturase, glycerol-3-phosphate, and acyltransferase, among other enzymes, play important roles in these processes [17].

In this study, Soxhlet extraction method and gas chromatography-mass spectrometry (GC-MS) were used to measure the oil content and composition changes with the development of seeds of *A. trifoliata*, and transcriptome sequencing was used to analyze the key genes of lipid biosynthesis. The main aims of this study were to (1) evaluate the changes in oil content and composition during the development of *A. trifoliata* seeds, and (2) determine the key FA biosynthesis genes in *A. trifoliata* seeds.

## 2. Materials and Methods

### 2.1. Plant Material

*A.**trifoliata* was cultivated in the experimental field of the Institute of Bast Fiber Crops, Chinese Academy of Agricultural Sciences (Yuanjiang, Hunan province). GD-3 was selected for study because of its lower ASO content and higher seed yield compared with other varieties in this area [12]. According to our statistics, the fruit weight of GD-3 is generally less than 200 g, and its peel is purple when ripe. GD3 maintains the biological characteristics of *A. trifoliata*; the fruit will crack when ripe and produces many seeds (more than 200). GD-3 has been planted for 4 years, and organic fertilizer was applied twice a year (5000 g/per plant) in March and October, respectively. The field management is in accordance with the normal *A. trifoliata* cultivation and management methods [18]. Flowers were simultaneously marked at the flowering stage, and fruits were harvested at 120, 135, 150, 165, 180, and 195 days after flowering (DAF) (F, S, K, T, U, and I). The seeds were separated from the pulp, cleaned with normal saline, and divided into two parts; one part was immediately frozen at −80 °C, and the other part was weighed and dried at low temperature. Three marked fruits were taken for each period. After all samples were collected, frozen samples were analyzed (F was frozen for 100 days, I was frozen for 5 days). The first sampling time was 6 July 2020, and the last sampling time was 16 September 2020. In July, the average daily temperature in Yuanjiang was 27–35 °C, and the total precipitation was 132 mm. In August and September, the average daily temperature in Yuanjiang was 25–32 °C and 21–28 °C and the total precipitation was 125 and 65 mm, respectively.

### 2.2. Dynamic Changes in ASO Content and FA Composition

Dried seeds (3 g) were selected from the samples in the same growth period. First, a small grinder was used to smash the seeds, and the resulting powder was filtered through a 30-mesh sieve. Then, the Soxhlet extraction method was used to determine ASO content [12]. The conditions for Soxhlet extraction were: temperature 50 °C, solid to liquid ratio 1:60, and extraction time 4 h. After the determination of ASO content, the sample was collected in a 1.5 mL centrifuge tube and stored at 4 °C. Finally, the FA composition of ASO was determined by GC-MS using an Agilent GC 7890 gas chromatograph and an Agilent 5977 mass spectrometer; helium (99.999% purity) was used as the carrier gas. The GC-MS operating conditions were as previously described by Sun et al. [19]. Before GC-MS analysis, FAs were converted to FA methyl esters (FAMEs). As described in previous studies [12,20], 0.06 g of ASO was diluted with diethyl ether/petroleum ether (1:1 *v*/*v*, 2 mL) and 0.4 M KOH-CH_3_OH (1 mL), vortexed, and maintained at room temperature (approximately 25 °C) for 2.5 h. Then, redistilled water (2 mL) was added to the mixture, which was then vortexed and centrifuged at 4500 rpm for 2 min. Finally, the organic phase containing FAMEs (100 mL) was collected and diluted with petroleum ether (900 mL).

### 2.3. cDNA Library Construction and Sequence Analysis and Alignment

Total RNA content was extracted from approximately 0.5 g of seed using the RNAprep Pure Plant kit (Tiangen, Beijing, China), and RNA concentration and purity were determined using a NanoDrop 2000 device (Thermo Fisher Scientific, Wilmington, DE, USA). Based on the analysis of oil content in developing *A. trifoliata* seeds, samples at five crucial stages (F, S, T, U, and I) were selected for transcriptomic analysis, and a total of 15 libraries were constructed for RNA-seq (each stage had three replicates). mRNA was purified from 1 μg of total RNA, fragmented, and then used to prepare a cDNA library using the NEBNext Ultra RNA Library Prep Kit (Illumina; NEB, Ipswich, MA, USA). cDNA library quality was assessed using the Agilent Bioanalyzer 2100 system (Agilent Technologies, Palo Alto, CA, USA). Illumina sequencing was performed using the HiSeq 2500 sequencing system. After the removal of reads containing poly-N and low-quality reads, the remaining clean reads were mapped to the reference *A. trifoliata* genome using HISAT2 or StringTie, from which unigenes were obtained [21,22].

### 2.4. Bioinformatic Analysis

P-Unigene expression levels were calculated as fragments per kilobase of exon model per million mapped fragments (FPKM) using the Cufflinks software package, and read counts for each gene were obtained using htseq-count. Gene expression levels in various samples were compared using the DESeq method, with *p* value < 0.05, fold-change > 2, or fold-change < 0.5, as thresholds indicating significant differences in gene expression [23]. The weighted gene co-expression network analysis (WGCNA) package was used to construct DEG co-expression networks [24]. A module containing at least 30 genes was constructed based on the scale-free network model. Then, an association analysis between the co-expression network and ASO, oleic acid (OA), and linoleic acid (LA) content was performed to screen for phenotype-associated modules. Gene Ontology and Kyoto Encyclopedia of Genes and Genomes pathway enrichment analyses of DEGs were performed using R, based on a hypergeometric distribution. Using the HMMER 3.1b software, the hidden Markov model was constructed based on the *Arabidopsis* and walnut *FAD* gene families and their *FAD* proteins were downloaded from GenBank (Appendix A). *A. trifoliata* protein sequences were obtained from our genomic database. ProtParam (https://web.expasy.org/protparam/ (accessed on 23 February 2022) was used to predict the physicochemical properties of the proteins, and Plant-mPLoc (http://www.csbio.sjtu.edu.cn/bioinf/plant-multi/# (accessed on 23 February 2022) was used to predict their subcellular localization.

### 2.5. Quantitative Analysis

Total RNA extraction was performed as described in Section 2.3. Approximately 0.5 μg of RNA and the PrimeScript RT Master Mix (Aidlab Biotechnologies, Co., Ltd., Changsha, China) were used to synthesize cDNA. A Bio-Rad CFX96 Touch detection system (Bio-Rad Laboratories, Richmond, CA, USA) and a SYBR Green PCR master mix (Aidlab Biotechnologies, Co., Ltd.) were used to conduct qPCR on each sample. We used EF-1α, which was found to be stably expressed in *A. trifoliata* [25], as an internal control gene. Primers for the qPCR experiments were designed using the Primer 5.0 software [26], and a total of nine lipid biosynthesis-related genes were analyzed. The qPCR system and procedures were developed based on the SYBR Green PCR Master Mix Kit (Aidlab Biotechnologies, Co., Ltd.) After PCR amplification, the Delta Ct method was used to analyze quantitative variations in each gene.

### 2.6. Statistical Analysis

Data (fruit weight, seed drying rate, ASO content, ASO composition, FPKM value, and gene expression levels) reported in the figures are averages of at least three different measurements. SAS 9.0 was used for one-way ANOVA based on Tukey’s test, and different letters represent significance at *p* ≤ 0.05.

## 3. Results

### 3.1. Dynamic Changes in A. trifoliata Fruit Weight and Seed Oil Content

At present, when *A. trifoliata* fruits crack naturally, they are considered mature (Figure 1A,B). In this study, 195 DAF samples were at this stage. Between periods F and K, *A. trifoliata* fruit weight increased by only 5.94 g; however, during this period, the seed drying rate increased from 26.81% to 49.82% (Figure 1C). Therefore, at this stage, fruits developed slowly while seeds developed rapidly. In contrast, from 60 to 90 DAF, fruits developed rapidly (fruit weight increased from 74.55 g to 137.51 g), and seeds developed slowly (no significant change in drying rate) (Figure 1C). At maturity, fruit weight was 203.63 g and seed-drying rate was 59.66%.

Figure 1D shows the dynamic changes in ASO content, which peaked (37.76% of seed weight) at 180 DAF; however, there was no significant change in its content between 135 and 165 DAF. There was a decreasing trend in its content between 180 and 195 DAF. ASO was found to contain 11 FA types, with the main FAs being palmitic acid (PA), stearic acid (SA), OA, and LA, which accounted for over 97% of ASO (Table 1), while the other seven FAs accounted for less than 1% of ASO. The FA types present in ASO did not change with seed development, and only their relative content changed. Among the four main FAs that constitute ASO, OA and LA exhibited the most significant changes (Figure 1E). LA content decreased from 37.91% to 28.17%, and with seed development, its relative content gradually decreased. OA content increased from 33.56% to 43.01%, and the trend in its change was opposite to that of LA. There was no significant change in PA content, and SA content increased from 2.62% to 4.75%.

In this study, the oil content was reflected by the ratio of oil in dry seeds. Therefore, combined with the dynamic change in the seed dry rate and oil content, the accumulation of seed oil was the highest at 180 DAF.

### 3.2. Transcriptome Sequencing

After the removal of reads containing poly-N and those of low quality, 108.45 GB clean reads were obtained from 15 cDNA libraries. A total of 44.34, 50.44, 50.83, 46.22, and 49.79 million clean reads were generated from the F, S, T, U, and I libraries, respectively. The GC content of the clean reads was 44.00%–44.78%, and 91.91%–94.83% Q30 bases (Appendix A). The clean reads were made freely available in the NCBI database (accession number: PRJNA79843).

Between 90.12% and 92.76% of clean reads were mapped to the reference *A. trifoliata* genome (unpublished data) (accession number: PRJNA750300), and 44,842 unigenes were identified from the transcriptome, 5399 of which were new unigenes that were not mapped to the genome. A total of 5399 new unigenes were annotated using the Basic Local Alignment Search Tool (BLAST). Searches were conducted against the NR, Swiss-Prot, GO, COG, KOG, Pfam, and KEGG databases, and 4143 new unigenes were annotated (Appendix A).

### 3.3. Analysis of DEGs

Through pairwise comparison of samples at each time point, 8756 DEGs were identified (Figure 2A) (Appendix A). To clarify the developmental mechanism of *A. trifoliata* seeds, we focused on DEG trends at different stages of seed development. Through WGCNA, changes in transcriptomic data were examined. Based on the scale-free network model, the soft threshold was set to 12 (Figure 2B), and 8756 DEGs were categorized into 10 modules (Figure 2C). The largest module was the light green module (4210 DEGs), and the grey module constituted a collection of genes that were not assigned to other modules (six DEGs) (Figure 2D). To better understand the relationship between the gene expression patterns of the modules and physiological traits, we conducted an association analysis. The tan (Appendix A), dark orange (Appendix A), and turquoise (Appendix A) modules were significantly related to FA, OA, and OA content (*p* ≤ 0.05) (Figure 2E), and these three modules constituted 2880 DEGs.

The FA content in I was lower than that in U, so we also focused on evaluating the 2296 DEGs between U and I. We performed GO and KEGG pathway enrichment analyses on DEGs obtained through WGCNA (three modules that were significantly related to physiological traits) and those in U and I. We focused on DEGs involved in pathways associated with the synthesis of plant oils, including FA biosynthesis, FA elongation, and triacylglycerol (TAG) biosynthesis. In these pathways, the DEGs of the three WGCNA modules were found to be significantly related to FA biosynthesis (Appendix A), while DEGs in U and I were found to be significantly related to FA elongation (Appendix A). Several genes involved in FA synthesis were identified. Figure 3 shows the ASO biosynthesis process (Table 2).

### 3.4. Identification of Genes Involved in FA Biosynthesis

The pyruvate dehydrogenase complex (*PDHC*), comprising four subunits (E1α, E1β, E2, and E3), is a rate-limiting enzyme that catalyzes the irreversible oxidative decarboxylation of pyruvate to acetyl-CoA, and acetyl-CoA is an FA synthesis precursor [27]. We identified only *PDH-E**1α* (Atr14G031810, Atr14G033430), *PDH-E1β* (Atr01G001230, Atr01G001270, Atr14G023850, Atr04G024990), and *PDH-E2* (Atr09G013640). *PDH-E1α* and *PDH-E1β* expression levels from U-I were higher than those from F-T, and this might explain why oil accumulates more from U-I (Table 2).

Acetyl-CoA carboxylases (*ACCases*) constitute a group of FA biosynthesis rate-limiting enzymes. *ACCases* consist of biotin carboxylase, the biotin carboxylase carrier protein (BCCP), α-carboxyl transferase, and β-carboxyl transferase [28], and previous research has shown that any of these subunits can influence lipid content [29]. We identified four genes encoding the *ACC-BCCP* subunits, and *Akebia*
*trifoliata*_newGene_7764 did not match the reference genome.

There are three 3-ketoacyl-ACP synthase (KAS) types in plants, and each type has a different function. *KASⅢ* catalyzes the synthesis of acetoacetyl-ACP; *KASⅠ* catalyzes the synthesis of 6–16 carbon compounds; and *KASⅡ* catalyzes the conversion of C16:0-ACP to C18:0-ACP [30,31]. Four genes encoding *KASII* and one gene encoding KASIII were identified. The expression levels of *KASII* and *KASIII* at the first stage were higher than those at the other four stages (Table 2).

### 3.5. Identification of Genes Involved in Unsaturated FA Biosynthesis

Over 70% of ASO constitutes unsaturated FAs, and *FAD* is a key enzyme in their synthesis [32,33]. In this study, we identified four *FAD* genes, including two *SAD* and two *FAD2* genes. *SAD* catalyzes the conversion of C18:0-ACP to C18:1-ACP, which is a key enzyme involved in the synthesis of C18:1 FAs. *FAD2* catalyzes the conversion of C18:1-ACP to C18:2-ACP, which is a key enzyme involved in the biosynthesis of C18:2 FAs. Twenty-three FAD gene family members were identified in the *A. trifoliata* genome based on the hidden Markov model. Through Pfam domain analysis, 20 *FAD* family genes were ultimately obtained and numbered according to their annotation in *A. trifoliata* (Table 3).

Subcellular localization analysis showed that these proteins were mainly located in the endoplasmic reticulum and chromosomes; *AtrFAD4*, *AtrFAD5*, and *AtrFAD7* were located in the cell membrane. Aside from *AtrFAD4* (298), *AtrFAD5* (295), and *AtrFAD8* (198), *AtrFAD* proteins consist of 300 to 470 amino acids. Using the neighbor-joining (NJ) method in the MEGA X software, *A. trifoliata FAD* protein sequences were constructed together with *Arabidopsis* and walnut *FAD* protein sequences to build a phylogenetic tree (Figure 4A). This indicated that these three species have similar *FAD* gene families. There were four main *FAD* subfamilies in *A. trifoliata*, the *SAD* desaturase subfamily, Δ7/Δ9 desaturase subfamily, Δ12/ω-3 desaturase subfamily, and the “front-end” desaturase subfamily, with most of the proteins being members of the *SAD* desaturase subfamily (seven). Interestingly, chromosome 7 was found to carry most *AtrFADs*, all of which were *SADs*. Four *AtrFADs (AtrFAD3*, *AtrFAD17, AtrFAD18*, and *AtrFAD19*) showed high expression levels (FPKM ≥ 100) (Figure 4B) (Table 3). The *AtrFAD17* and *AtrFAD18* levels observed could explain why *A. trifoliata* seed oil had a higher unsaturated FA content during the early stages of seed development (Figure 4B).

### 3.6. Identification of Genes Involved in TAG Biosynthesis

Glycerol-3-phosphate acyltransferase (*GPAT*) is the most important key enzyme involved in TAG biosynthesis, and it catalyzes the acylation of glycerol-3-phosphate (G-3-P) sn1 [34]. In *Arabidopsis*, *GPAT* is in the endoplasmic reticulum or the plastid (ATS1), with plastid *GPAT* being soluble [35]. We identified six *GPAT* and two *ATS1* types in *A. trifoliata*. As shown in Table 2, the expression levels of *GPAT* and *ATS1* were different, with those of *ATS1* being lower than those of *GPAT*.

Diacylglycerol acyltransferase (*DGAT*) is the rate-limiting enzyme of the TAG biosynthesis process. *DGAT* catalyzes the conversion of 1,2-diacylgycerol to TAG; this step is regarded as the key step in TAG synthesis by the Kennedy pathway [36]. Phospholipid:diacylglycerol acyltransferase (PDAT) is another enzyme involved in TAG synthesis [37]. It catalyzes the transfer of the FA in phosphatidylcholine to diphenol glycerol to produce lysophosphatidylcholine and TAG. Only one *DGAT1* and two *PDAT* types were identified in *A. trifoliata* in this study; *DGAT1* and *PDAT* showed similar expression trends, with their expression levels being higher from U-I (Table 2). From stages U-I, *DGAT1* expression levels were higher than those of *PDAT*. This may also indicate that the Kennedy pathway is the main pathway for TAG biosynthesis.

### 3.7. qPCR Analysis of Lipid-Related Genes

Nine key genes involved in FA biosynthesis were randomly selected and evaluated using the qRT-PCR method. Figure 5A–I shows the expression levels of these genes at five different stages. Each graph shows the changes in the expression levels of each gene as determined by qRT-PCR and RNA-Seq. Figure 5 shows that the trends in expression levels as determined by qRT-PCR and RNA-Seq were highly similar, indicating that the expression data obtained by RNA-Seq was reliable.

## 4. Discussion

Fossil fuels are some of the most important substances in the world but are mainly concentrated in specific geographic areas [38]. Plant oil has been considered a substitute for fossil fuels. Previous studies have showed that some plant oils could be used to produce biodiesel [39,40]. Although the present study showed that the highest oil content of GD-3 was 37.76%, we previously reported that the highest oil content of 130 *A. trifoliata* germplasms was 51.27%, with an average of 43.44% [13]. This discrepancy may be due to differences in the varieties, periods, or the main components of ASO. Nonetheless, *A. trifoliata* seed oil has a high application value as either biodiesel or edible oil. As research on ASO is ongoing, we should first focus on the process for producing biodiesel from ASO in the near future.

When the fruit of *A. trifoliata* naturally cracks, it is at the mature stage [41]. However, the present study demonstrated that 15 days before natural fruit cracking (180 DAF), the dry rate and oil content of *A. trifoliata* seeds were significantly higher than those in the cracking period (195 DAF). Therefore, when *A. trifoliata* is used as an oil crop, it should be harvested earlier. Previous studies have shown that the oil content of plant seeds is due to the dynamics between oil biosynthesis and oil degradation. When the grain is mature, its oil biosynthesis rate decreases and oil degradation increases, which would lead to a certain degree of decline in oil content in the mature and later stages [42]. However, a limitation of the present study was that we only analyzed one *A. trifoliata* variety and growing environment, so our conclusion that the harvesting period of *A. trifoliata* seeds should be earlier needs to be verified in follow-up experiments.

Plant seed oil content is a complex quantitative trait that is regulated by multiple genes. Previous reports have shown that genes such as *PDHC* complex, *ACCase* complex, *KAS*, *FAD*, *DGAT*, and *LPAAT* play important roles in the regulation of lipid content and composition [27,29,32,37]. The *PDHC* complex regulates the biosynthesis of acetyl-CoA, the precursor of FA synthesis. *PDHC* is composed of four subunits, the activities of which affect the synthesis of acetyl-CoA, which in turn affects the content of plant oil [27]. In the present study, we did not identify the PDH-E3 subunit, but after functional retrieval of all DEGs in the transcriptome, we found the PDH-E3 subunit and subunit-related genes that were not identified as *ACCase*. The reason for this phenomenon may be that WGCNA compresses the number of DEGs used for candidate gene mining. Fatty acid elongation 1 (*FAE1*) is a type of 3-ketoacyl-CoA synthase (KCS) found in higher plants. KCS catalyzes the first step of the very long chain FA (VLCFA) biosynthesis process [43,44]. Edible oils with high VLCFA content are regarded as being of poor quality [45,46]. Therefore, in many oil crop breeding programs, VLCFA content is reduced to improve the nutritional value of the oil. We only found approximately 0.7% of C20:0 and C20:1 in ASO, and this was similar to their content in 130 germplasms as determined by Zhong et al. [12]. VLCFA content in ASO was low, but we identified many genes associated with *FAE1* in *A. trifoliata*. However, the expression level of these genes was not high (Table 2), which may explain why, even with the high number of *FAE1* genes, VLCFA content was very low.

## 5. Conclusions

*A. trifoliata* seed and fruit development were not synchronized. Between stages A-K and T-U, *A. trifoliata* seeds developed rapidly but fruits developed slowly. When the fruits were ripe, their average weight was 203.63 g and seed-drying rate was 59.66%. Seeds had the highest oil content during the U period, and with further fruit ripening, the oil content and seed drying rate showed a decreasing trend. Therefore, for *A. trifoliata* used as an oil crop, its seeds need to be harvested in advance prior to full ripening. Relative LA content was highest during the F period and gradually decreased with seed development, but OA showed the opposite trend. As the seeds developed, relative OA and LA content changed from 33.56% to 43.01% and from 37.91% to 43.01%, respectively. In addition, there was no significant change in PA content (from 23.77% to 22.89%); the relative content of SA was less than 5%, changing from 2.62% to 4.75% (highest) as the seeds developed. RNA-Seq results showed that there were 8756 DEGs in the different comparison groups, and that between F and S, and S and T, there were only 417 and 210 DEGs, respectively. Through WGCNA analysis, the 8756 DEGs were divided into 10 different modules, of which three, which contained 2880 DEGs, were significantly related to phenotype. KEGG and GO analyses showed that these 2880 DEGs were enriched in FA-related processes and pathways. FAD gene family analysis showed that there were 20 AtrFAD family members in *A. trifoliata*, and these could be divided into four sub-groups. Several specific genes related to FA and TAG biosynthesis, including *ACC-BCCP*, *PDH-E2, FAD2*, and *SAD* were identified. These findings provide a basis for ASO development and *A. trifoliata* breeding.

## Figures and Tables

**Figure 1 biology-11-00855-f001:**
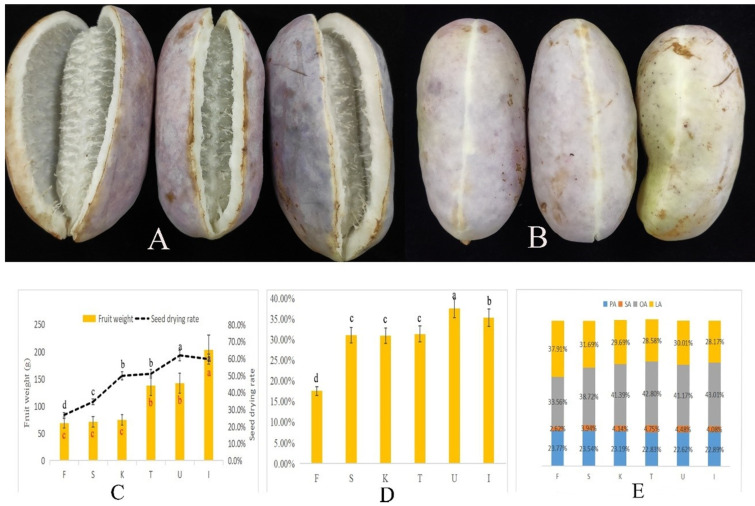
Morphological characteristics and oil content during *A. trifoliata* seed development. (**A**) Ripe fruits. (**B**) Unripe fruits. (**C**) Dynamic changes in fruit weight and seed drying rate. (**D**) Dynamic changes in ASO content. (**E**) Dynamic changes in the four main ASO components. The different letters (a, b, c...) represent significance at *p* ≤ 0.05.

**Figure 2 biology-11-00855-f002:**
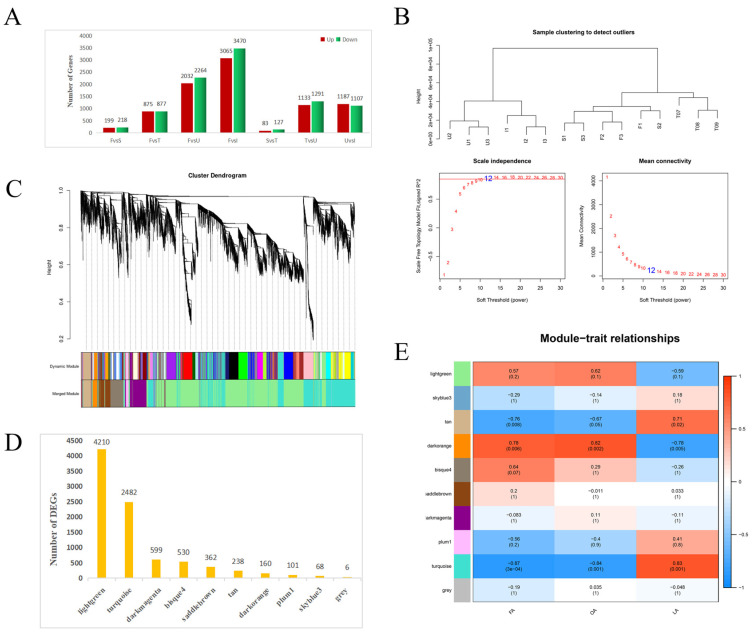
WGCNA of DEGs. (**A**) DEGs identified by pairwise comparison between samples at each time point. (**B**) Power of the WGCNA. (**C**) WGCNA cluster dendrogram. (**D**) DEG number in each module. (**E**) Correlation of the expression patterns of the modules to physiological traits.

**Figure 3 biology-11-00855-f003:**
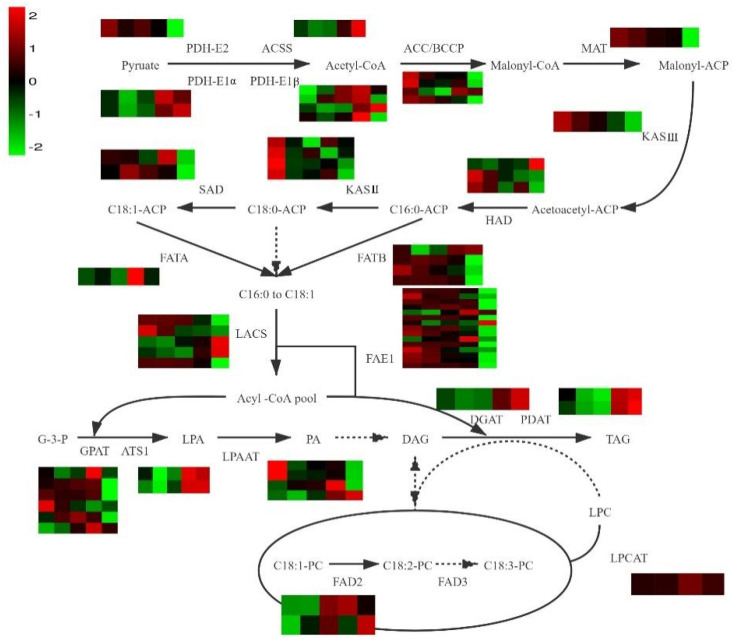
Lipid biosynthesis transcriptional model in developing *A. trifoliata seeds*. The five squares in each row represent the five developmental stages; each square in each column represents a gene; red and green represent high and low expression levels, respectively.

**Figure 4 biology-11-00855-f004:**
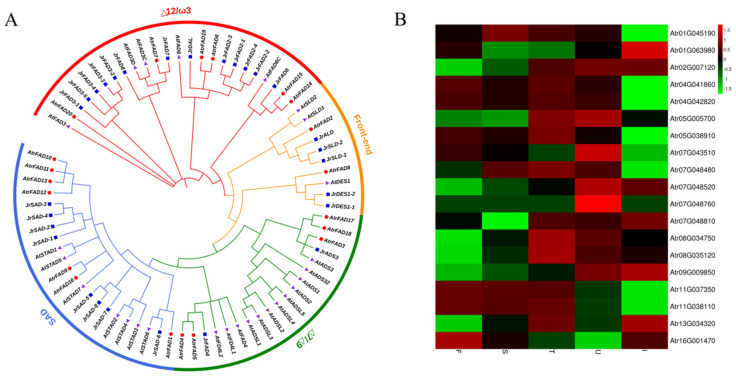
Putative fatty acid desaturase (FAD) unigenes identified in *A. trifoliata*. (**A**) Phylogenetic analysis of *JrFADs* (*Juglans regia* L.), *AtFADs* (*Arabidopsis thaliana*), and *AtrFADs* (*A. trifoliata*) using nucleotide sequences. (**B**) Heat map of *AtrFADs* based on FPKM.

**Figure 5 biology-11-00855-f005:**
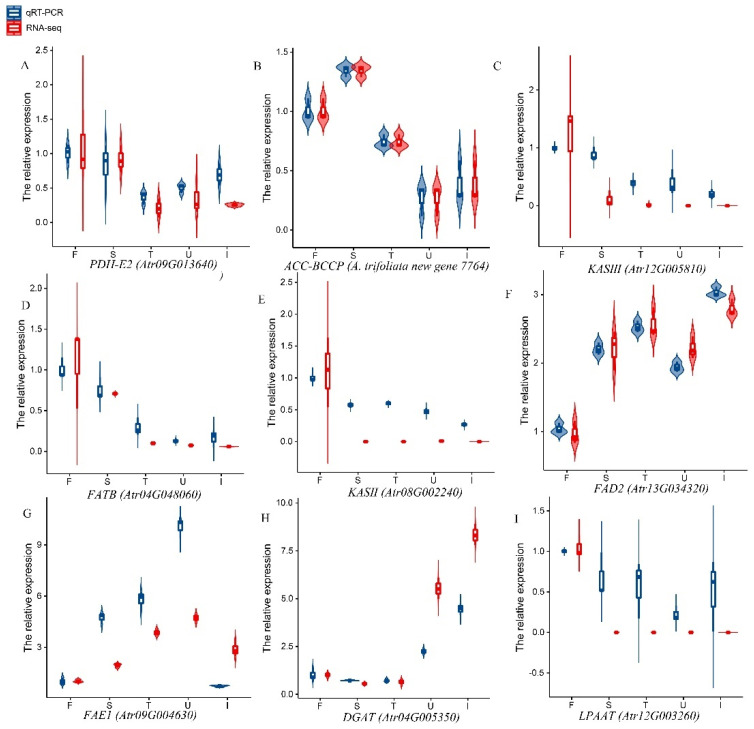
qRT-PCR analysis of nine important genes associated with lipid biosynthesis. The expression of all genes was normalized: blue and red represent qRT-PCR and RNA-seq expression levels, respectively. (**A**) The relative expression level of *PDH-E2*. (**B**) The relative erpression level of *ACC-BCCP*. (C) The relative erpression level of *KAS**Ⅲ*. (**D**) The relative erpression level of *FATB*. (**E**) The relative erpression level of *KAS**Ⅱ*. (**F**) The relative erpression level of *FAD2*. (**G**) The relative erpression level of *FAE1*. (**H**) The relative erpression level of *DGAT*. (**I**) The relative erpression level of *LPAAT*.

**Table 1 biology-11-00855-t001:** Average content of ASO composition in developing *A. trifoliata* seeds. Different letters represent significance at *p* ≤ 0.05.

Compositions (%)	F	S	K	T	U	I
C14:0	0.20 ab	0.20 ab	0.21 a	0.18 b	0.19 b	0.21 a
C16:0	23.77 a	23.54 a	23.19 ab	22.83 b	22.62 b	22.89 b
C16:1	0.48 a	0.48 a	0.45 b	0.31 d	0.40 c	0.43 b
C17:0	0.20 a	0.18 b	0.13 c	0.10 d	0.13 c	0.14 c
C17:1	0.12 a	0.12 a	0.10 b	0.10 b	0.10 b	0.00 c
C18:0	2.62 d	3.94 c	4.14 b	4.75 a	4.48 ab	4.08 bc
C18:1	33.56 d	38.72 c	41.39 b	42.80 a	41.17 b	43.01 a
C18:2	37.91 a	31.69 b	29.69 c	28.58 d	30.01 c	28.17 d
C18:3	0.57 a	0.55 a	0.30 c	0.19 d	0.30 c	0.39 b
C20:0	0.33 b	0.37 a	0.21 c	0.10 d	0.32 b	0.36 a
C20:1	0.26 b	0.21 c	0.19 d	0.06 e	0.29 b	0.33 a

**Table 2 biology-11-00855-t002:** Identification of lipid-related genes in developing seeds.

Enzyme	KEGG Annotation	Gene ID	Gene Expression Level				
			F	S	T	U	I
PDH-E1α	pyruvate dehydrogenase E1 component alpha subunit [EC:1.2.4.1]	Atr14G031810	34.82	20.58	31.76	86.20	59.54
		Atr14G033430	12.17	5.83	8.96	29.54	49.31
PDH-E1β	pyruvate dehydrogenase E1 component beta subunit [EC:1.2.4.1]	Atr14G023850	32.10	22.99	25.20	56.18	81.96
		Atr01G001270	4.77	9.94	11.60	13.63	5.10
		Atr04G024990	1.01	0.81	1.01	5.61	0.60
PDH-E2	pyruvate dehydrogenase E2 component (dihydrolipoamide acetyltransferase) [EC:2.3.1.12]	Atr09G013640	7.42	5.97	6.56	5.65	3.20
ACC-BCCP	acetyl-CoA carboxylase biotin carboxyl carrier protein	Atr14G034290	41.19	31.02	27.76	28.30	16.91
		Atr15G033160	49.04	26.98	25.63	32.15	15.65
		Atr01G010810	10.40	6.40	4.89	11.74	10.35
		Akebia_trifoliata_newGene_7764	7.59	6.92	6.06	5.03	3.14
MAT	[acyl-carrier-protein] S-malonyltransferase [EC:2.3.1.39]	Atr05G032500	45.07	41.11	36.37	34.81	17.76
KASⅢ	3-oxoacyl-[acyl-carrier-protein] synthase III [EC:2.3.1.180]	Atr12G005810	18.53	13.90	11.71	8.65	5.49
HAD	3-hydroxyacyl-[acyl-carrier-protein] dehydratase [EC:4.2.1.59]	Atr08G008530	34.10	29.97	37.20	34.65	89.15
		Atr11G026150	35.59	27.84	22.55	19.28	18.18
		Atr01G047740	27.63	24.27	16.63	24.09	14.17
KASⅡ	3-oxoacyl-[acyl-carrier-protein] synthase II [EC:2.3.1.179]	Atr03G013000	12.65	7.05	3.73	9.51	7.33
		Atr07G009450	14.85	7.62	9.82	5.37	8.21
		Atr03G057450	39.15	25.09	25.48	27.23	21.91
		Atr08G002240	31.60	18.59	20.02	23.37	13.56
SAD	acyl-[acyl-carrier-protein] desaturase [EC:1.14.19.2 1.14.19.11 1.14.19.26]	Atr07G043510	529.94	502.14	374.17	814.09	246.83
		Atr01G045190	83.93	139.64	108.93	90.12	36.20
FAD2	acyl-lipid omega-6 desaturase (Delta-12 desaturase) [EC:1.14.19.6 1.14.19.22]	Atr13G034320	232.53	417.51	635.19	371.35	873.55
		Atr05G005700	3.11	3.04	8.00	9.05	5.50
FATB	fatty acyl-ACP thioesterase B [EC:3.1.2.14 3.1.2.21]	Atr08G012910	107.49	114.40	109.96	97.92	21.44
		Atr08G012370	75.74	88.06	81.30	64.73	19.41
		Atr04G048060	3.61	1.66	0.93	0.52	0.09
		Atr02G007540	0.68	0.25	0.40	0.83	0.95
FATA	fatty acyl-ACP thioesterase A [EC:3.1.2.14]	Atr07G024450	53.72	60.74	49.22	110.67	58.98
LACS	long-chain acyl-CoA synthetase [EC:6.2.1.3]	Atr06G006720	36.88	33.60	48.22	55.27	100.21
		Atr11G013800	12.03	11.79	11.32	6.73	2.95
		Atr07G001260	38.88	41.14	37.52	22.96	2.22
		Atr11G034050	7.41	3.01	1.00	0.87	0.56
		Atr02G017850	0.98	0.84	0.74	1.41	2.65
FAE1	3-ketoacyl-CoA synthase [EC:2.3.1.199]	Atr04G006760	96.46	83.59	80.09	34.99	0.40
		Atr06G016990	46.74	26.00	35.06	9.40	1.79
		Atr07G006560	40.24	11.52	3.94	1.80	0.05
		Atr01G055960	17.47	13.52	13.99	10.67	9.15
		Atr02G000850	26.95	14.79	10.00	15.93	0.06
		Atr02G061280	0.14	0.19	0.22	0.17	0.03
		Atr03G011460	1.44	1.09	2.29	4.91	3.27
		Atr04G047630	6.22	4.79	5.55	5.05	8.56
		Atr01G019190	1.28	0.92	0.94	0.63	0.04
		Atr09G004630	1.66	3.77	4.71	23.40	1.21
		Atr15G034500	37.47	37.03	36.44	35.01	23.21
		Atr05G047000	23.22	1.99	0.23	0.02	0.01
		Atr02G056530	8.14	5.77	3.63	2.87	0.60
		Atr02G056560	4.51	3.67	1.80	1.76	0.27
		Atr06G010840	14.00	11.02	11.46	9.99	1.23
GPAT	glycerol-3-phosphate O-acyltransferase 1/2 [EC:2.3.1.15]	Atr09G000930	8.65	11.90	15.20	17.83	0.36
		Atr12G002270	36.13	28.21	28.59	35.24	4.56
		Atr16G001970	9.05	2.97	1.60	0.88	1.35
		Atr05G014350	42.35	64.68	83.48	60.68	19.91
		Atr05G039970	0.07	0.01	0.03	1.77	0.02
		Atr05G043400	0.01	0.04	0.46	6.38	0.52
ATS1	glycerol-3-phosphate O-acyltransferase [EC:2.3.1.15]	Atr03G046480	3.00	2.17	2.59	4.34	4.07
		Atr03G047060	5.11	4.56	5.49	7.37	7.39
LPAAT	lysocardiolipin and lysophospholipid acyltransferase [EC:2.3.1.-2.3.1.51]	Atr05G010470	54.78	36.01	39.86	41.25	30.69
		Atr05G010480	41.01	26.86	27.45	29.32	21.23
		Atr12G003260	6.94	7.88	8.03	10.26	5.91
		Atr11G034300	3.22	2.19	4.17	9.08	14.07
DGAT1	diacylglycerol O-acyltransferase 1 [EC:2.3.1.20 2.3.1.75 2.3.1.76]	Atr04G050350	2.11	1.24	1.45	9.27	25.77
PDAT	phospholipid:diacylglycerol acyltransferase [EC:2.3.1.158]	Atr04G004940	3.51	1.73	1.42	7.18	8.24
		Atr04G005010	2.77	1.93	1.70	5.98	7.71
LPCAT	lysophospholipid acyltransferase [EC:2.3.1.51 2.3.1.23 2.3.1.-]	Atr03G002590	4.97	5.08	8.66	5.75	0.60

**Table 3 biology-11-00855-t003:** Information on AtrFADs.

Name	Subfamilies	Chromosome	Number of Amino Acids	Subcellular Location	Gene Expression Level				
					F	S	T	U	I
*AtrFAD1*	SAD	Chr 1	385	Chloroplast	77.90	152.03	108.93	90.12	14.58
*AtrFAD2*	Front-end	Chr 1	446	Endoplasmic reticulum	28.73	13.65	15.16	25.22	59.98
*AtrFAD3*	Δ7/Δ9	Chr 2	384	Chloroplast, Endoplasmic reticulum	4.67	7.86	16.08	18.82	18.38
*AtrFAD4*	Δ7/Δ9	Chr 4	298	Cell membrane, Cell wall, Chloroplast, Mitochondrion	0.14	0.02	0.16	0.03	0.00
*AtrFAD5*	Δ7/Δ9	Chr 4	295	Cell membrane, Cell wall, Cell nucleus	0.34	0.07	0.47	0.19	0.00
*AtrFAD6*	Δ12/ω3	Chr 5	397	Endoplasmic reticulum	3.24	3.00	8.00	9.05	4.92
*AtrFAD7*	Δ12/ω3	Chr 5	457	Cell membrane, Endoplasmic reticulum	22.62	20.10	26.67	18.92	7.95
*AtrFAD8*	Front-end	Chr 6	195	Chloroplast, Endoplasmic reticulum	0.00	0.00	0.00	0.00	0.00
*AtrFAD9*	SAD	Chr 7	397	Chloroplast	528.17	461.17	374.17	814.09	260.38
*AtrFAD10*	SAD	Chr 7	397	Chloroplast	1.36	2.73	3.36	2.60	0.52
*AtrFAD11*	SAD	Chr 7	395	Chloroplast	0.11	0.34	0.69	4.65	1.94
*AtrFAD12*	SAD	Chr 7	336	Chloroplast	0.00	0.00	0.00	0.03	0.00
*AtrFAD13*	SAD	Chr 7	467	Chloroplast	0.01	0.00	0.35	0.17	1.67
*AtrFAD14*	Δ12/ω3	Chr 8	347	Chloroplast	8.63	13.51	19.96	17.33	14.14
*AtrFAD15*	Δ12/ω3	Chr 8	341	Chloroplast	7.94	11.55	17.80	14.74	13.46
*AtrFAD16*	SAD	Chr 9	397	Chloroplast	47.63	59.75	68.32	95.17	106.35
*AtrFAD17*	Δ7/Δ9	Chr 11	322	Chloroplast, Endoplasmic reticulum	47,497.02	36,879.01	33,223.82	2311.01	87.18
*AtrFAD18*	Δ7/Δ9	Chr 11	332	Chloroplast, Endoplasmic reticulum	41,957.81	31,117.12	38,430.20	2062.99	99.40
*AtrFAD19*	Δ12/ω3	Chr 13	381	Endoplasmic reticulum	228.41	414.97	635.19	371.35	758.35
*AtrFAD20*	Δ12/ω3	Chr 16	461	Chloroplast, Endoplasmic reticulum	1.63	0.99	0.70	0.42	1.18

## Data Availability

The materials of this study were provided by the Institute of Bast Fiber Crops, Chinese Academy of Agricultural Sciences. Correspondence and requests for materials should be addressed to Mingbao Luan (luanmingbao@caas.cn). The sequencing data have been deposited in NCBI SRA database (accession number: PRJNA79843).

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
