# Peer review of "Transcriptome Analysis and GC-MS Profiling of Key Fatty Acid Biosynthesis Genes in *Akebia trifoliata* (Thunb.) Koidz Seeds"

_biology, 2022, doi:10.3390/biology11060855_

Round 1
Reviewer 1 Report
The manuscript needs rigorous revision in the English language.
The discussion part is missing.
Address all the comments raised in the PDF file.
can not be accepted in its present form.

Author Response
Thank you for your suggestion and all the comments raised in the PDF file have been addressed, and the changed parts were written in blue. The discussion has been added in line 304-343.

Reviewer 2 Report
To the Authors (in detail):
1) In the title and in the whole manuscript: to avoid confusion, please use the correct and updated botanical nomenclature, for example according to www.gbif.org, and also report the authorship and (in brackets) the botanical family in the title and at the first mention. In the rest of the text it is possible to indicate the species as Akebia trifoliata or A. trifoliata;
2) Simple Summary, line 13, please, verify and correct the scientific name;
3) Simple Summary, line 14, please, verify and correct the scientific name and insert one space before .. However;
4) Simple Summary, lines 15, 20 and in the whole manuscript, please use the International binomial nomenclature: the scientific name have to be italicized;
5) Introduction section, lines 44-47. This is an important part of the introduction section and has to be extended and supported with other references because the vegetable oils available as alternative to petroleum. Please, find, read and discuss also [1-3]:
[1] Seed oil from ten Algerian peanut landraces for edible use and biodiesel production.
J. Oleo Sci. 65 (1) 9-20 (2016). DOI: 10.5650/jos.ess15199
[2] Tomato seed oil: a comparison of extraction systems and solvents on its biodiesel and edible properties.
[3] Influence of high temperature and duration of heating on the sunflower seed oil properties for food use and bio-diesel production.
J. Oleo Sci. 66, (11) 1193-1205 (2017). DOI: 10.5650/jos.ess17109
6) Introduction section, line 51: unsuitable for which consumption?
7) Introduction section, lines 52-53, verify how to write the scientific names;
8) Introduction section, line 65: Soxhlet in capital letter;
9) Introduction section, lines 73-74-75, please, verify the English form: to analysis the oil content?
10) 2.1 sub-section, when you have written the scientific names you have used different criterion, for example, here you have not insert one space before the species;
11) 2.1 sub-section, please, describe fruits and seeds for an International reader. Include some biometric data;
12) 2.1 sub-section: no information is given about agronomic procedures: climate; soil; fertilizers (type, quantity, period); irrigation (y/n); pest treatments;
13) 2.1 sub-section: how long between -80 °C and analysis?
14) 2.1 sub-section: -80 °C is a refrigerator?
15) 2.2 sub-section: solid to liquid ratio: 1 g seeds and 60 mL solvent? Why this ratio?
16) 2.2 sub-section, line 99 and in the whole manuscript, insert one space before the first bracket;
17) 2.2 sub-section: fatty acids were obtained from FAMEs? Or it is the contrary?
18) 2.3, 2.5 sub-section:, lines 110, 144, the scientific name italicized;
19) 2.6 sub-section: have you applied one-way ANOVA? Please, specify. Indicate also the applied test;
20) 3.1 sub-section, lines 180-183: this is true in your environmental and agronomic condition, please, include this. You cannot affirm that this is always true;
21) Figure 1D, in the y axis it is not necessary to include the decimals after comma (delete .00);
22) Figure 1C, y axis, do not include decimals (.0 is not necessary);
23) Table 1, insert the % symbol only after Compositions and delete it after each fatty acid;
24) Table 1: include the one-way ANOVA in each line to compare different variables;
25) Table 1, in the caption, the scientific names have to be italicized;
26) Figure 1 caption and in the whole manuscript, you have used different criteria to write the significance, in particular the spacing between letter, symbol and numeric value, please, verify and be consistent in the whole manuscript and tables and figures;
27) Captions of figures and table, please, verify the template and some recently published paper for the punctuation after the table or figure number. Please, be consistent with the instructions for authors of Biology;
28) References section: the bibliography has to be re*-arranged as per Biology Instructions for authors;
29) References section, line 429, please verify if this scientific name is correct;
30) References section: the scientific names have to be italicized, please, see the original titles of papers you have listed;
31) Please, write in blue color or evidence differently the corrections you will do.
I suggest a major revision
Regards.
Author Response
2.5 Introduction section, lines 44-47. This is an important part of the introduction section and has to be extended and supported with other references because the vegetable oils available as alternative to petroleum. Please, find, read and discuss also [1-3]:
Response:
Thank you for your suggestion and these references have been added in discussion (line 306-307)
2.6 Introduction section, line 51: unsuitable for which consumption?
Response:
Thank you for your suggestion and this sentence has been changed to ‘.it will be considered unsuitable as edible oil’ line 53.
2.13 2.1 sub-section: how long between -80 °C and analysis?
Response:
Thank you for your suggestion and the samples for RNA-seq and qRT-PCR were placed at 80 °C refrigerator about 80 days
2.15 2.2 sub-section: solid to liquid ratio: 1 g seeds and 60 mL solvent? Why this ratio?
Response:
Thank you for your suggestion. In another study, we have optimized the Soxhlet extraction method for A. trifoliata seeds The results showed that the optimum process conditions was 50 °C, 4h and 1:60. (Zhong, Y., Zhang, Z., Chen, J., Niu, J., Shi, Y., Wang, Y., Chen, T., Sun, Z., Chen, J., Luan, M. Physicochemical properties, content, composition and partial least squares models of A. trifoliata seeds oil. Food Chem X. 2021, 12: 100131. doi: 10.1016/j.fochx.2021.100131.)
2.17 2.2 sub-section: fatty acids were obtained from FAMEs? Or it is the contrary
Response:
Thank you for your suggestion. The sentence has been changed to ‘fatty acids need to be converted to fatty acid methyl esters (FAMEs)’ in line 112
2.19 2.6 sub-section: have you applied one-way ANOVA? Please, specify. Indicate also the applied test;
Response:
Thank you for your suggestion. In fact, the significance in this article is based on the results of one-way ANOVA, but we didn't make it clear, and we have added it to line 167.
2.20 3.1 sub-section, lines 180-183: this is true in your environmental and agronomic condition, please, include this. You cannot affirm that this is always true
Response:
Thank you for your suggestion. Although there were study showed that harvest time would affect palnt oil content, but just based on one variety and environmental can prove that this is true. So the viewpoint has been moved to disscussion, and changed to ‘But in this study, we just chose a A. trifoliata variety and growing environment, this viewpoint (the harvesting period of A. trifoliata seed should be earlier) need be be proved by follow-up experiments’(line 324)
Other suggestion.
Response:
Thank you for your suggestion. All suggestion have been revised, and we have written all the changes in blue. Thank for your great help in improving the scientificity of this paper.
Round 2
Reviewer 1 Report
The manuscript can be accepted.
Some minor improvements can be done to the English language.
Author Response
Dear Reviewer
Thank you for your help to make this paper more scientific. This paper has been revised by professional English Editing Agency, We hope it can meet the requirements of Biology-basel.
Reviewer 2 Report
To the Authors (in detail):
- the argument is interesting but it has to be improved. The authors have included some of my comment, anyway they have to improve more. The M&M section has to be better detailed. Inaccuracies in the manuscript. The references section is not written as suggested by Biology Instructions for Authors.
- 1 sub-section, page 3, line 93: samples were frozen at -80 °C and not refrigerated;
- 1 sub-section: for how long samples were frozen (-80 °C) before analysis? Write it in the manuscript;
- 6 sub-section: you have you applied one-way ANOVA. Please, specify also the applied test (Tukey or what?)
- 6 sub-section, line 164, please, delete: (a, b, c, ..);
- 4 sub-section and in the whole manuscript, please, verify in the template how to insert tables and figures in the text;
- section, line 301: biodiesel;
- Section, line 357 and in the whole manuscript: the scientific names have to be italicized as per the International binomial nomenclature;
- Caption of table 1: do not write colons after the table number but a dot;
- Captions of figures: for example: Figure 1. And not Fig 1. Please be consistent with the template and also verify with some recently published papers;
- The captions of tables and figures have not to be written in bold. Please be consistent with the template and also verify with some recently published papers
- References section: the bibliography has to be re-arranged as per Biology Instructions for authors, there is a lot of mistakes. The journal name has to be abbreviated and italicized; the year of publication in bold; the volume number has to be italicized; the punctuation is wrong; the issue number is not required. There is more than one mistake for each line. Please, verify carefully and indent each reference;
- Ref 13 and in the whole section: the initial letter of each word has to be in capital letter;
- Ref 39, the correct abbreviation is: Ital. Sostanze Gr.
- References section, ref 25, please verify if this scientific name is correct (Akebia trifoliata and not trifoliate);
- References section: the scientific names have to be italicized, please, see the original titles of papers you have listed;
- Please, write in red color or evidence differently the corrections you will do.
I suggest a major revision
Regards.
Author Response
Dear Reviewer,
Thank for your great help in improving the scientificity of this paper.
At present, there are some papers that will also get Akebia trifoliata writeen as Akebia trifoliate (such as 10.1016/j.ecolind.2020.106093,10.1016/j.jiec.2021.06.009),but most of paper write it as Akebia trifoliata. In the manuscript, we just put the table and figure in the end because they are too large.
All suggestion have been revised, and we have written all the changes in red.
With kindest regards,
Yours Sincerely,
Mingbao Luan
Round 3
Reviewer 2 Report
Manuscript Number: biology-1702364, titled:
Transcriptome analysis and GC-MS profiling of key fatty acid biosynthesis genes in Akebia trifoliata seeds.
Review 3 – 17 May 2022
Dear Editor of Biology
the argument is interesting and well treated. The authors have included my comments.
I suggest the publication of this manuscript in the present form.
Regards.
Author Response

(The authors gave the same response as above.)
